# Effects of Different Production Methods on the Quality and Microbial Diversity of Sauerkraut in Northeast China

**DOI:** 10.3390/foods13233947

**Published:** 2024-12-06

**Authors:** Weichao Liu, Yunchao Wang, Tong Zhao, Yunfang Zheng, Guangqing Mu, Fang Qian

**Affiliations:** 1School of Food Science, Dalian Polytechnic University, Dalian 116034, China; 221710832000966@xy.dlpu.edu.cn (W.L.); jinsedehupo@163.com (Y.W.); 231710832001016@xy.dlpu.edu.cn (Y.Z.); mugq@dlpu.edu.cn (G.M.); 2Dalian Center for Certification and Food and Drug Control, Dalian 116021, China; icp8300@sina.com

**Keywords:** northeast sauerkraut, physicochemical properties, microbial diversity, volatile components, flavour, correlation analysis

## Abstract

Sauerkraut is a popular fermented food in Northeast China. However, owing to the different production methods used, the quality of commercial sauerkraut is often quite different, which is reflected mainly in the differences between starter culture (Group-L), additive addition (Group-P) and natural fermentation (Group-H) methods. The purpose of this study was to explore the differences among the three fermentation methods by measuring physical and chemical indices, microbial diversity indices, flavour indices and volatile substances. The results revealed that there was no significant difference in the physical or chemical indices among the groups. The content of esters and alcohols in Group-L was the highest, and the taste richness, aftertaste-a and aftertaste-b were the highest, which had a positive effect on flavour. The highest level of microbial diversity was found in Group-H, which contained many pathogenic bacteria, such as *Janibacter*, *Pseudomonas*, and *Vagococcus*, which reduced the food safety of sauerkraut. At the genus level, the dominant bacterial genera in the starter and additive groups included *Lactobacillus* and *Pediococcus*. The correlation analysis revealed that Group-L was positively correlated with the contents of *Lactobacillus plantarum*, *Lactobacillus brevis*, *Pediococcus*, ethyl oleate and vanillin. In summary, this study evaluated the different production methods of northeast sauerkraut, providing theoretical support for the production of high-quality northeast sauerkraut.

## 1. Introduction

Northeast sauerkraut is a common speciality food in Northeast China. It is made by fermenting cabbage in salt and lactic acid by microorganisms, which extends the shelf-life of fresh vegetables. Sauerkraut not only retains the original nutritional components of vegetables but also endows vegetables with a variety of colours, aromas and tastes, improves taste, and facilitates digestion [1,2]. The regular consumption of sauerkraut can reduce cholesterol, prevent cardiovascular and cerebrovascular diseases, increase beneficial bacteria in the gastrointestinal tract, prevent constipation, etc. [3,4].

Northeast Chinese cabbage, with its thick petiole and high water content, is prone to rot during fermentation, so fermented vegetables are usually used to extend the preservation time [5]. Many factors affect the quality of sauerkraut during fermentation and preservation, such as the salt concentration, pH, temperature, fermentation mode, etc. [6]. The rapid reduction in pH in the early stage of fermentation is crucial to the quality of sauerkraut because it can minimise the activity of spoilage bacteria and even directly inhibit or kill spoilage bacteria. The preservation time of sauerkraut at different temperatures also varies. Many studies have shown that the quality of sauerkraut fermented in autumn and winter is significantly better than that fermented in summer, which can inhibit the growth of spoilage bacteria and pathogens to a certain extent [7,8]. Among these influencing factors, the fermentation method has the most extensive influence on the flavour, safety, microbial community and taste of sauerkraut.

Natural fermentation is inevitably affected by many factors, such as the season of production, the amount of salt used, hygienic conditions of operation, etc., making rot more likely to occur. In addition, the use of natural fermentation also has many drawbacks, such as a long fermentation cycle, unstable fermentation quality, inability to reach factory scale, and standardised production. Adding food additives such as sodium D-isoascorbate and sodium metabisulfite to sauerkraut can effectively prevent the occurrence of rot; adding vitamin C and sodium dehydroacetate can block the generation of nitrite [2], reduce its content, and prevent sauerkraut from mildew. Reducing rancidity and odour [1]. However, food additives should be used within the scope of addition. If excessive sodium pyrosulfite is used as an example, sulphur dioxide will remain in the sauerkraut, reducing the quality and safety of the sauerkraut [3]. Fermentation with starters can control fermentation conditions, accelerate fermentation, shorten the fermentation period, and improve the quality and safety of sauerkraut [9]. However, compared with the naturally fermented sauerkraut, the different strains contained in the starter culture may affect the production of the characteristic flavour of sauerkraut. At present, studies on sauerkraut have focused mostly on the influence of a certain fermentation agent on sauerkraut flavour and other indicators, as well as the influence of environmental conditions during fermentation on various indicators of sauerkraut [10,11]; however, few studies have compared starter fermentation, additive fermentation and natural fermentation methods used in the production of northeast sauerkraut [12,13].

In this study, sauerkraut samples produced via different fermentation methods were purchased from Northeast China, and quality evaluations of physical and chemical indices, flavour indices and microbial diversity analysis were carried out to reveal the influence of fermentation conditions on the production of sauerkraut flavour substances and the production of the main microbial communities. Correlation analysis was conducted between different flavour substances and microbial communities. The present methods for the production of sauerkraut were evaluated scientifically to provide theoretical support for the industrial production of sauerkraut.

## 2. Materials and Methods

### 2.1. Sauerkraut Samples

A total of 15 different brands of sauerkraut were purchased. According to their production methods, they were divided into lactic acid bacteria fermentation group Group-L (LYY, LZY, LHX, LTQ, LYT), additive group Group-P (PZX, PLE, PBCC, PLW, PBC), and farmer-made group Group-H (HHB, HXM1, HXM2, HCY, HDL). An appropriate amount of sauerkraut from each group was homogenised and stored at −20 °C for subsequent determination of microbial diversity. The remaining samples were stored at 4 °C until use.

### 2.2. Determination of Physical and Chemical Indices

The pH was measured via a pH meter (PHS-25; Shanghai Mettler Company, Shanghai, China). The total acid content was assessed using acid–base titration methods [14]. Following a 15 min boiling of the sample in water, the pH was adjusted to 8.2 ± 0.2 using a 0.05 mol/L sodium hydroxide solution (Damao Chemistry; Tianjin, China). The volume of sodium hydroxide consumed during the titration was documented, and the total acid (TA) content was subsequently calculated based on the concentration of lactic acid expressed in grams per 100 millilitres.

The nitrite content was determined via colourimetry [15]. After the sample was incubated in a water bath for 15 min, it was mixed with saturated borax, and 106 g/L potassium ferrocyanide (Aladdin, Los Angeles, CA, USA) and 220 g/L zinc acetate (Aladdin, Los Angeles, CA, USA) were added to the solution and mixed evenly. After standing, the supernatant was mixed with 2 mL of 4 g/L p-aminobenzenesulfonic acid (Aladdin, Los Angeles, CA, USA) solution and 1 mL of 2 g/L N-(1-naphthalene) ethylenediamine hydrochloric acid (Aladdin, Los Angeles, CA, USA) for 20 min, and the absorbance was measured at a wavelength of 538 nm. The nitrite content was calculated on the basis of a standard curve. The nitrite content was expressed in mg/kg.

### 2.3. GC-MS Analysis

The volatile constituents of each sauerkraut sample were analysed using an Agilent 6890/5975 gas chromatography–mass spectrometry (GC-MS) system (Agilent Technologies, Santa Clara, CA, USA). Take 5 g of homogenised sauerkraut sample, place it in a headspace bottle, and place it in a water bath at 55 °C for 30 min. After preheating, insert the SPME (solid-phase microextraction) fibre head and completely absorb it for 30 min, and then insert it into the GC injector at 250 °C for 5 min. For the gas chromatography (GC) analysis, a nonpolar column (DB-35MS UI, 30 m × 0.32 mm ID × 0.25 μm) was employed, with the injection port maintained at 220 °C. The temperature program commenced at 50 °C for a duration of 1 min, followed by a ramping phase where the temperature increased at a rate of 8 °C/min until it reached 180 °C, which was held for an additional minute. Subsequently, the temperature was elevated to 280 °C at a rate of 15 °C/min and maintained for 10 min. The interface temperature was set to 280 °C, with a flow rate of 1.5 mL/min and a solvent delay of 2.5 min. For the polar column analysis, a DB-WAX UI column (30 m × 0.25 mm ID × 0.25 μm) was utilised, also with an injection port temperature of 220 °C. The temperature program initiated at 50 °C for 1 min, increased to 180 °C at a rate of 8 °C/min, and then further escalated to 240 °C at a rate of 15 °C/min for a duration of 10 min. The interface was maintained at 250 °C, with a flow rate of 1.0 mL/min and a solvent delay of 3.2 min. Tridecanoic acid methyl ester, at a concentration of 10 μg/mL, was utilised as an external standard for quantitative analysis.

### 2.4. Microbial Diversity Analysis

Microbial diversity in frozen sauerkraut samples (−20 °C) was assessed via the E.Z.N.A.^®^ Soil DNA Kit (Omega Biotek, Norcross, GA, USA). The extraction of total genomic DNA was subsequently followed by quality assessment using 1% agarose gel electrophoresis, along with quantification and purity analysis conducted with a NanoDrop2000 (Thermo Scientific, Waltham, MA, USA). The polymerase chain reaction (PCR) amplification targeting the V3-V4 region of the 16S rRNA gene was performed in two stages: the initial PCR utilised Barcode-tagged primers 799F (5′-AACMGGATTAGATACCCKG-3′) and 1392R (5′-ACGGGCGTGTGTRC-3′), while the secondary PCR employed primers 799F (5′-AACMGGATTAGATACCCKG-3′) and 1193R (5′-ACGTCATACCCCACCTTCC-3′). The amplification protocol included predenaturation at 95 °C for 3 min, 27 cycles (95 °C denaturation, 55 °C annealing, and 72 °C extension, each for 30 s), a final extension at 72 °C for 10 min, and storage at 4 °C. Postamplification purification was performed with an AxyPrep DNA Gel Extraction Kit (Axygen Biosciences, Union City, CA, USA), and the products were assessed via 2% agarose gel electrophoresis and quantified with a Quantus™ fluorometer (Promega, Madison, WI, USA). Purified PCR products were prepared for sequencing via the NEXTFLEX Rapid DNA-Seq Kit (Bioo Scientific, Austin, TX, USA) and sequenced via Illumina MiSeq PE300/NovaSeq PE250 platform (USA).

Quality control of the paired-end raw sequencing data was performed via fastp (v. 0.19.6, https://github.com/OpenGene/fastp, accessed on 23 December 2022). Paired-end reads were merged with FLASH (v. 1.2.11, http://www.cbcb.umd.edu/software/flash, accessed on 23 December 2022). OTU (operational taxonomic unit) clustering and chimaera removal were conducted with UPARSE (v. 7.1, http://drive5.com/uparse/, accessed on 23 December 2022) on the basis of a 97% similarity threshold for sequence alignment.

### 2.5. Electronic Tongue Assay

Electronic tongue detection was conducted with a TS-5000Z taste analysis system (Insent, Atsugi, Japan). The sensor was primed for 24 h with an internal solution (3.3 mol/L KCl and saturated AgCl) and a reference solution (30 mol/L KCl and 0.3 mol/L tartaric acid). The system quantified various taste attributes—including sourness, bitterness, astringency, aftertastes b and a, umami, tanginess, and saltiness—of the samples at ambient temperature. For reliability, each sample underwent quadruplicate analysis with the electronic tongue.

### 2.6. Electronic Nose Assay

The flavour of sauerkraut was determined via the electronic nose (Airsense, Schwerin, DE) method [16]. The sauerkraut sample was placed in a headspace bottle at 37 °C for 30 min to allow the flavour components to completely dissipate. During the measurement, the probe and air needle of the e-nose were inserted at the same time. After the sensor was self-cleaned for 60 s, the measurement was performed for 60 s, and the data analysis was conducted mainly between 59 and 60 s.

### 2.7. Statistical Analysis

Each sample underwent triplicate parallel measurements, and the resultant data are presented as means ± standard deviations (x ± s). Statistical analyses were conducted using SPSS version 19.0 (SPSS, Inc., Chicago, IL, USA), employing one-way ANOVA to determine significant differences at a significance threshold of *p* < 0.05. Data visualisation was performed using Origin (2021b Origin Lab, Inc., Northampton, MA, USA).

## 3. Results and Discussion

### 3.1. Physicochemical Characteristics of Sauerkraut

Table 1 shows that the pH of the sauerkraut samples was between 3.99 and 5.0, and the total acid content was between 1.281 g/100 g and 2.191 g/100 g. There was no significant difference among the three groups of sauerkraut samples, but there was a significant difference among the different sauerkraut samples. The content of nitrite was low, and no nitrite was detected in the lactic acid bacteria group, additive group or homemade group, which was much lower than the 20 mg/kg level in the Chinese national standard. As each group of sauerkraut is a regular commercial sauerkraut, physical and chemical indicators are regulated, so no significant difference is foreseeable. However, as long as homemade sauerkraut is fermented and mature, the content of nitrite can still reach the standard of safe consumption, but the nitrite reduction effect is poor, the time required is longer, and its quality cannot be guaranteed. Previous studies have also reported that the nitrite content of pickled vegetables made by farmers exceeds 20 mg/kg [17], so controlling the nitrite content is still very important.

### 3.2. Analysis of Volatile Components

The GC-MS method was used to determine the flavour substances associated with the different methods used to produce fermented sauerkraut. A total of 60 volatile components, including 18 acids, 2 aldehydes, 5 hydrocarbons, 1 sulphide, 15 esters, 1 nitrile, and 7 alcohols, were detected in all sauerkraut samples. A total of 41, 21 and 25 aroma components were found in the Group-L, Group-P and Group-H groups, respectively, and the types of volatile components in Group-L were significantly greater than those in the other groups. In terms of all the volatile components, the content of acid compounds was the highest in all the groups, and the contents of aldehydes and sulphides were the lowest; however, because of the high odour thresholds of aldehydes and sulphides, they could also affect the flavour at relatively low concentrations [18]. A heatmap (Figure 1A) revealed that the content of esters in the Group-L group was greater than that in the other groups and that the acid content in the Group-P group was significantly greater than that in the other groups. Most flavour substances are widely found in previously fermented vegetables [19,20]. Acids, sulphides and esters greatly contribute to the aroma of fermented vegetables [21]. In all the sauerkraut samples, the content of acid volatile components, especially dehydroacetic acid, in the Group-P group was significantly greater than that in the other groups. Esters are the main volatile components of cruciferous plants. This compound is considered to have a fruit flavour [22]. These volatile components are characteristic flavour substances of spicy cabbage. The higher the content of esters is, the better the sensory quality of sauerkraut [23]. The contents of esters and alcohols in the Group-L group were the highest and were significantly greater than those in the other groups (*p* < 0.05). The reason for the increase in alcohols in the Group-L group may be the corresponding increase in esters due to the increase in acids and alcohols, and lactic acid bacterial fermentation can promote this process [24].

The volatile components of each group of sauerkraut were analysed by OPLS-DA, and the substances with VIP values greater than 1 and *p* < 0.05 were usually regarded as the iconic differentially abundant metabolites between groups [25]. Figure 1C shows that the different metabolites are ethyl oleate, 2-meme 3-butanediol, ethyl 9,12,15-octadecatrienoate, dehydroacetic acid, vanillin, phenylethyl alcohol, 9,12-octadecadienoic acid, butanoic acid, and 9,12,15-octadecatrienoic acid (Z, Z, Z, Z). The score chart (Figure 1B) shows that there were differences in volatiles among the three groups.

### 3.3. α Diversity Analysis

Illumina HiSeq sequencing yielded a total of 709,189 raw reads from 15 samples. Following the removal of low-quality data, 627,943 high-quality labels were retained, with a mean length of approximately 426 base pairs. At a sequence similarity threshold of 97%, the coverage exceeded 99% (refer to Table 2), suggesting that the sequence length was adequate for the objectives of this study. Furthermore, as the sequence length increased, the Chao index exhibited a decreasing trend (see Figure 2A), while the Shannon index remained stable (see Figure 2B), indicating a sufficiently high recognition coverage of bacterial taxa. The alpha diversity of the sauerkraut bacterial community is presented in Table 2, revealing that the number of operational taxonomic units (OTUs) ranged from 33 to 323. The Chao1 index and Shannon index can reflect the richness and diversity of bacterial community OTUs, respectively. Figure 2C,D show that the OTU richness and diversity of Group-H are greater than those of the other groups, which is the same as the characteristics of naturally fermented sauerkraut OTUs reported in other studies [26] and significantly differs from those of the other groups. Group-P is the lowest, the Chao1 index is between 31 and 145, and the Shannon index is between 0.98 and 2.07.

Among all sauerkraut samples, Group-H had the richest microbial community, indicating that most bacterial growth was not restricted, which also increased the risk of sauerkraut being contaminated by microorganisms. The species and quantity of Group-P bacteria were low, indicating that the growth of bacteria was well inhibited and that the rot of sauerkraut was prevented.

### 3.4. Microbial Community Structure Analysis

The bacterial composition identified in all sauerkraut samples encompassed 19 phyla, 37 classes, 96 orders, 176 families, 304 genera, and 408 species. The analysis of the bacterial community structure was conducted at both the phylum and genus levels. At the phylum level (Figure 3A), the predominant bacterial groups included Firmicutes, Bacteroidota, Cyanobacteria, Proteobacteria, and Actinobacteria. The main dominant bacteria in each group was Firmicutes, and the same conclusion was reached in other studies [27]. Firmicutes was the most common bacteria in Group-P, with a relative abundance of more than 90%. In Group-L, the LYT sample showed that Firmicutes was not the dominant phylum, and the abundance of Firmicutes in the LZY group was only 50%. In Group-H, the abundance of Firmicutes was approximately 80%, and the number of species at the phylum level was obviously greater than that in the other two groups. At the genus level (Figure 3B), the bacteria with high abundance included *Lactobacillus*, *Pediococcus*, *Prevotella* and *Rhodococcus*. The relative content of *Lactobacillus* in Group-H was significantly lower than that in the other two groups. *Lactobacillus* are the main bacteria in the fermentation process of sauerkraut, which are related to the production of organic acids and the formation of flavour substances in sauerkraut. Its abundance has a significant impact on the quality of sauerkraut [28]. Lactic acid bacteria are directly added to sauerkraut as a fermentation agent, which directly increases the abundance of *Lactobacillus*, while the addition of additives may indirectly increase the abundance of *Lactobacillus* by reducing the abundance of harmful bacteria.

The LDA diagram (Figure 4A) revealed that the characteristic bacteria in Group-H were *Janibacter*, *Pseudoalteromonas*, *Enterococcus*, *Vagococcus*, *Pseudomonas* and others. *Janibacter* belongs to Actinobacteria. *Janibacter* can cause vaginal infection, respiratory tract infection, and even severe bacteraemia or septicaemia [29]. *Pseudomonas* is a genus of harmful intestinal bacteria, and intestinal colonisation by *Pseudomonas* is associated with an increased risk of pulmonary infection and mortality [30]. The characteristic microbial communities of Group-P included Firmicutes, Lactobacillales, and Leuconostocaceae. There is no characteristic population of Group-L. The characteristic populations of each group indicate that there are many conditional pathogenic bacteria in the characteristic genus of Group-H, and the microbial safety of sauerkraut produced without inoculation and fermentation and without additives is low, which is disadvantageous to the industrial production of sauerkraut.

The level of OTUs among the three groups of sauerkraut can be seen from the PCoA diagram (Figure 4B). There was little difference among the samples in Group-L, but there was a great difference between Group-P and Group-H; thus, the microbial diversity of sauerkraut fermentation can be controlled by the addition of lactic acid bacteria.

### 3.5. Flavour Evaluation

Figure 5 shows that the response values of R6, R9, R7 and R2 are relatively high, indicating that alkanes, aromatics, sulphur compounds and nitrogen oxides strongly contribute to flavour. There was no significant difference in flavour composition among the sauerkraut samples (*p* > 0.05), but the content was different. Among them, the content of flavour substances in Group-L was greater than that in the other two groups. Aromatics and alkanes are beneficial to the flavour of sauerkraut, and their high content indicates that sauerkraut has a better flavour [10]. Notably, for the R8 sensor (alcohols, aldehydes and ketone compounds), the response value of Group-L is also greater than that of the other groups, which is consistent with the conclusion drawn from the volatile component determination that Group-L contains more alcohol substances. There was no significant difference in flavour content between Group-P and Group-H, but the sulphide content of the additive group was lower than that of the other groups, and inorganic sulphides usually caused sauerkraut to smell poorly, which indicated that the flavour of homemade sauerkraut was worse than that of Group-L and Group-P.

The electronic tongue determination results for different sauerkraut samples are shown in Table 3. Except for those of the LZY samples, the fresh substances of Group-L were greater than 1.20, and those of the other two groups were less than 0.8. The flavour richness of Group-L was greater than 1.25. The Group-L values of aftertaste-a and aftertaste-b were also higher than those of the other two groups. The above indices revealed that the flavour of Group-L was better than that of Group-H and Group-P.

### 3.6. Correlations Between Microbial Diversity and Volatile Components

Volatile compounds are important components of the special flavour of sauerkraut. In addition to the varieties of cabbage, the types of microorganisms also have a great influence on flavour substances [31]. There is a strong correlation between *Lactobacillus* and the flavour substances of sauerkraut, and the same conclusion was reached in previous studies [32].

To explore the effects of different fermentation methods on the volatile flavour of sauerkraut, canonical correlation analysis (CCA) was performed on the differential microorganisms between the groups in the above analysis and the differential volatile flavour substances. The correlation between microbial diversity and flavour substances was analysed via CCA. [22]. The correlation coefficient diagram (Figure 6A) revealed that *Phyllobacterium myrsinacearum* was significantly positively correlated with 1,3-propanediol and hexadecanoate acid propyl ester; *Weissella* was significantly positively correlated with vanillin and hexadecanoate acid propyl ester; and *Lactobacillus brevis* was significantly positively correlated with ethyl oleate. This may be due to the unique metabolic processes of these bacteria, which produce substances that play a positive role in the flavour of sauerkraut. Some of these strains have also reached the same conclusion in previous studies [33,34]. The significant correlations among *Lactobacillus plantarum*, *Lactobacillus brevis* and *Pediococcus* also indicate that there may be synergistic effects among the three strains on the growth of cabbage.

Canonical correlation analysis (CCA, Figure 6B) revealed that the Group-L group was positively correlated with *Lactobacillus plantarum*, *Lactobacillus brevis*, *Pediococcus*, ethyl oleate and vanillin, which was consistent with the high content of esters in lactic acid bacteria mentioned earlier. There was a positive correlation between the Group-P group and *Lactobacillus sakei* and dehydroacetic acid. Dehydroacetic acid was the most abundant acid in Group-P. It is a substance with broad-spectrum antimicrobial effects. Its large presence may lead to a decrease in the diversity of Group-P microorganisms, which in turn affects the flavour of Group-P fermented sauerkraut, resulting in its poor flavour. The bacteriocin sakacin P produced by *Lactobacillus sakei* is a very promising preservative, which can also explain why *Lactobacillus sakei* can be widely found in sauerkraut with additives [25]. On the other hand, the distance between the samples in Group-H is large, and the clustering result is poor, which precisely shows that the quality gap between the sauerkraut samples in Group-H is large and that the quality is unstable.

## 4. Conclusions

This study explored the effects of different fermentation methods on the quality of Northeast China’s sauerkraut by conducting physical and chemical analyses, microbial diversity analysis, and flavour and volatile substance analysis on sauerkraut produced via different production methods in Northeast China. The sauerkraut fermented with the starter culture had the most types of volatile substances, among which esters and alcohols had the highest contents and the highest taste richness, aftertaste-a and aftertaste-b, indicating that the sauerkraut in the fermented with starter culture group had a better flavour, whereas the flavour in the additive group was the worst. From the perspective of microbial diversity, Group-L and Group-P inhibited the growth of a variety of pathogens, such as *Janibacter*, *Pseudomonas*, *Vagococcus*, etc., so the bacterial diversity was low, and the main genera were *Lactobacillus* and *Pediococcus*. Through correlation analysis, Group-L was found to be positively correlated with *Lactobacillus plantarum*, *Lactobacillus brevis*, *Pediococcus*, ethyl oleate and vanillin. This study helps to better control the quality of Northeast China’s sauerkraut, thus laying the foundation for the production of higher-quality Northeast China’s sauerkraut.

## Figures and Tables

**Figure 1 foods-13-03947-f001:**
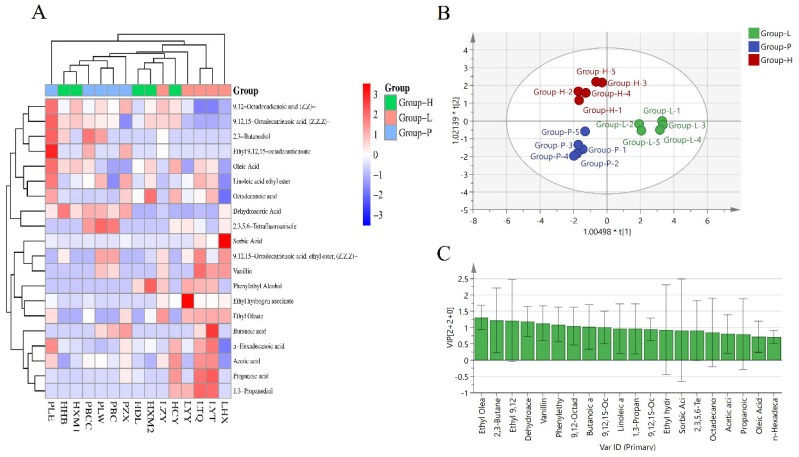
Vegetable flavour components of samples with different sauerkraut heatmaps: OPLS-DA score (**A**) graph (**B**) and variable importance in projection (VIP) figure (**C**).

**Figure 2 foods-13-03947-f002:**
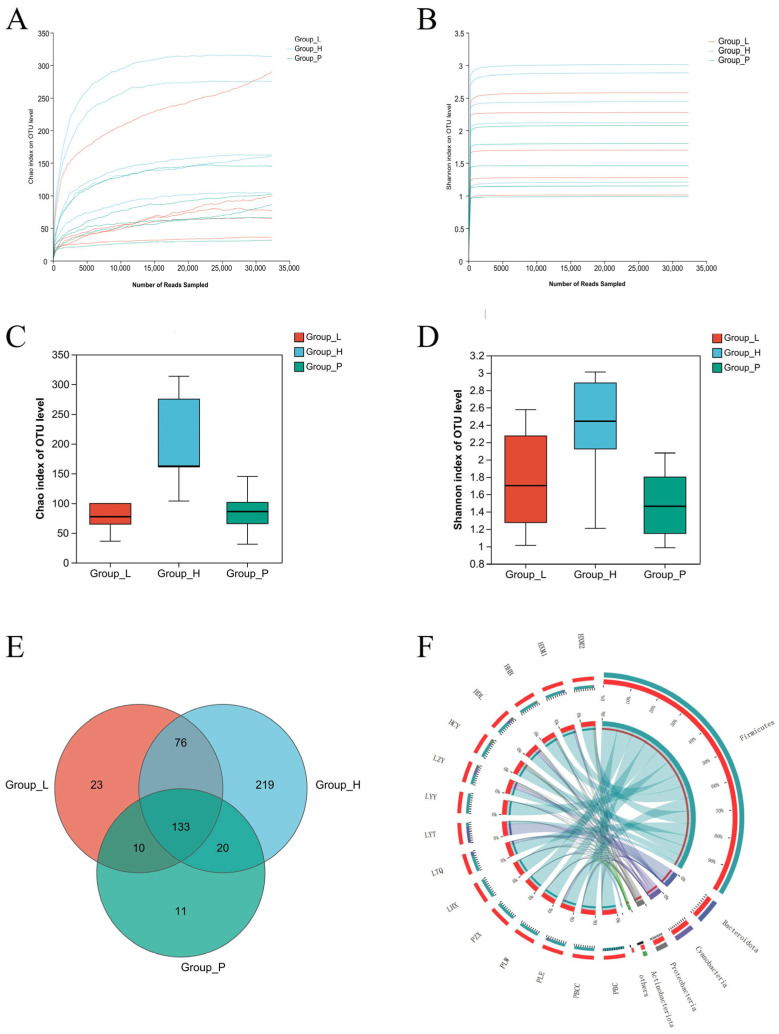
Chao1 index dilution curve (**A**) and Shannon index dilution curve (**B**), Chao1 index box plot (**C**), Shannon index box plot (**D**), Venn diagram (**E**) at the OTU level and Circos diagram (**F**) of Sauerkraut samples from each group.

**Figure 3 foods-13-03947-f003:**
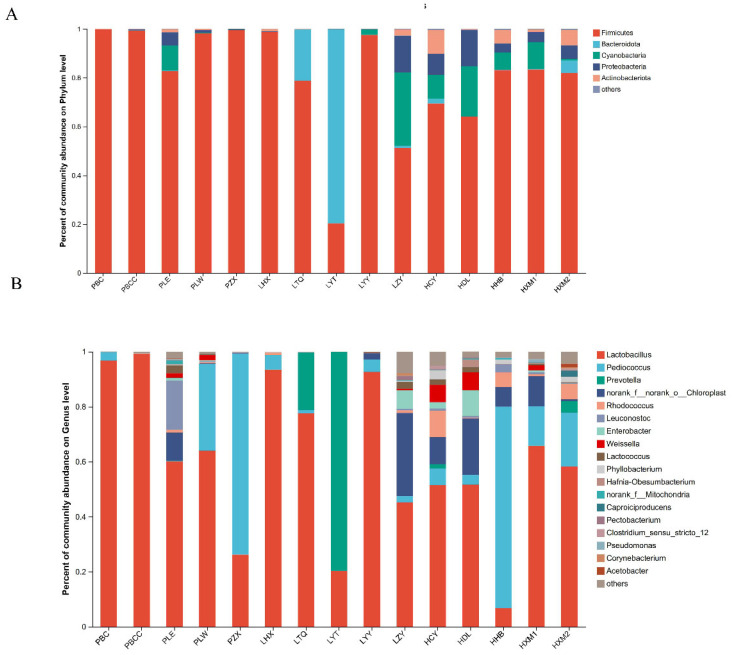
Relative abundance of bacterial communities at the phylum (**A**) and genus (**B**) levels in Sauerkraut samples from each group. Genera with relatively high relative abundances appear in the top 20.

**Figure 4 foods-13-03947-f004:**
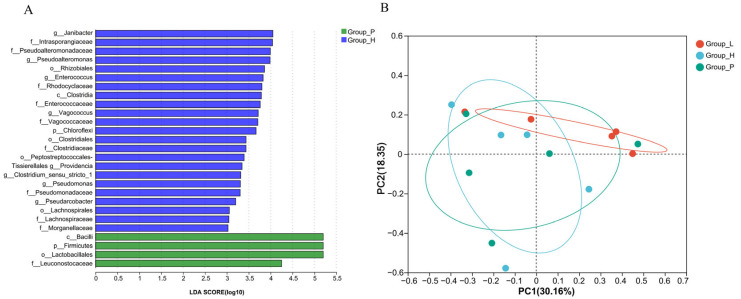
LDA discriminant diagram (**A**) and PCoA diagram at the OTU level of sauerkraut samples in each group(**B**).

**Figure 5 foods-13-03947-f005:**
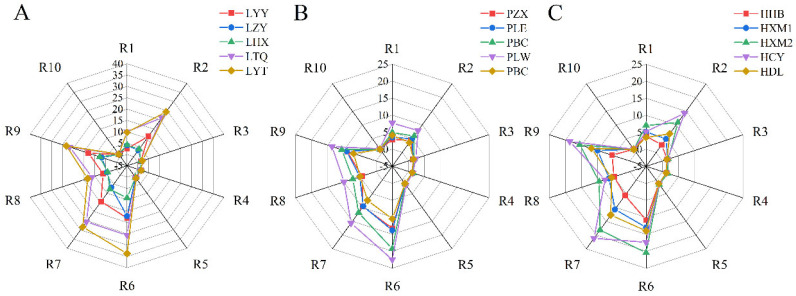
Electronic nose radar map of commercially available sauerkraut samples: Group-L (**A**), Group-P (**B**), and Group-H (**C**).

**Figure 6 foods-13-03947-f006:**
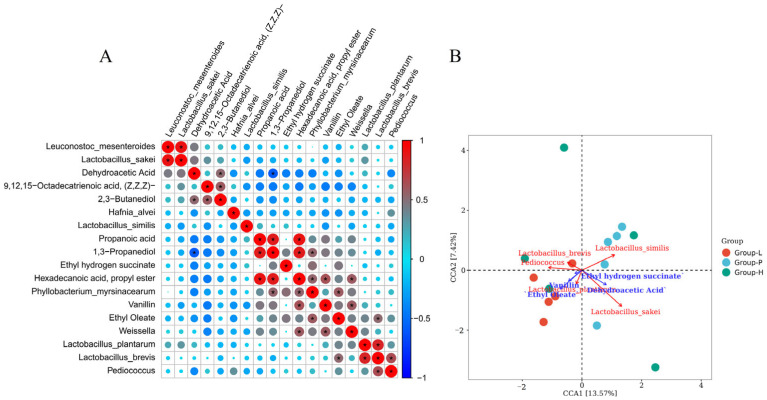
Microbial diversity of different samples of sauerkraut and the correlation coefficient of flavour substances (**A**) and CCA (canonical correlation analysis) chart (**B**). * Indicates significant correlation (*p* < 0.05).

**Table 1 foods-13-03947-t001:** Physicochemical indices of the sauerkraut samples in each group.

	pH	TA (g/100 g)	Nitrite (mg/kg)
LYY	4.09 ± 0.006 ^hi^	1.834 ± 0.024 ^c^	ND
LZY	4.20 ± 0.021 ^f^	1.532 ± 0.011 ^g^	ND
LHX	3.99 ± 0.006 ^k^	1.920 ± 0.025 ^b^	ND
LTQ	4.90 ± 0.010 ^b^	1.785 ± 0.043 ^cd^	ND
LYT	5.00 ± 0.058 ^a^	1.639 ± 0.017 ^f^	ND
PZX	4.44 ± 0.006 ^d^	1.630 ± 0.060 ^f^	ND
PLE	4.07 ± 0.006 ^ij^	2.113 ± 0.026 ^a^	ND
PBCC	4.11 ± 0.023 ^gh^	1.985 ± 0.062 ^b^	ND
PLW	4.18 ± 0.012 ^g^	2.121 ± 0.013 ^a^	ND
PBC	4.06 ± 0.010 ^ij^	1.727 ± 0.023d ^e^	ND
HHB	4.02 ± 0.020 ^jk^	1.281 ± 0.061 ^i^	ND
HXM1	4.04 ± 0.006 ^ij^	1.933 ± 0.051 ^b^	ND
HXM2	4.32 ± 0.010 ^e^	1.649 ± 0.051 ^ef^	ND
HCY	4.59 ± 0.055 ^c^	2.191 ± 0.066 ^a^	ND
HDL	4.41 ± 0.010 ^d^	1.435 ± 0.005 ^h^	ND

Different letters in Tables a~k indicate significant differences within the same column (*p* < 0.05).

**Table 2 foods-13-03947-t002:** Statistical table of α diversity analysis of sauerkraut samples in each group.

Sample	Ace	Chao	Coverage	Shannon	Simpson	Sobs
LHX	140.564	77.25	0.999351	1.275733	0.43705	51
LTQ	128.7289	99.75	0.999413	2.27447	0.166488	57
LYT	37.58594	36	0.999846	1.013059	0.6325	31
LYY	85.61422	65	0.999537	1.699681	0.265214	50
LZY	262.5409	290	0.998424	2.577678	0.151196	215
PBC	33.08772	31.33333	0.999846	1.46291	0.346568	28
PBCC	66.93092	65.8	0.999598	0.985399	0.521537	58
PLE	152.0301	145.2609	0.999166	2.076664	0.190257	130
PLW	126.3552	101.6	0.999259	1.800983	0.24561	74
PZX	71.23272	86	0.999475	1.148852	0.506839	52
HCY	282.9555	275.0833	0.998733	3.010269	0.080329	258
HDL	113.3075	103.8667	0.999289	2.122533	0.187438	87
HHB	169.6204	162.3448	0.99895	1.208238	0.548007	143
HXM1	158.1996	161.0588	0.99898	2.44315	0.1431	130
HXM2	323.1312	313.5424	0.998579	2.883016	0.103772	296

**Table 3 foods-13-03947-t003:** Electronic tongue result statistics of commercially available sauerkraut samples.

Sample Name	Sour	Bitter	Astringent	Aftertaste-B	Aftertaste-A	Umami	Richness	Salinity
LYY	−3.44 ± 0.01 ^f^	0.47 ± 0.01 ^efg^	−1.29 ± 0.11 ^d^	−0.77 ± 0.04 ^e^	0.83 ± 0.05 ^abc^	1.54 ± 0.18 ^ab^	1.52 ± 0.28 ^a^	−0.31 ± 0.01 ^g^
LZY	−3.23 ± 0.06 ^ef^	1.21 ± 0.01 ^d^	0.11 ± 0.03 ^bc^	1.26 ± 0.14 ^a^	0.35 ± 0.05 ^d^	0.56 ± 0.01 ^cd^	0.69 ± 0.01 ^c^	1.15 ± 0.04 ^f^
LHX	−4.05 ± 0.05 ^g^	0.56 ± 0.05 ^ef^	−1.29 ± 0.04 ^d^	1.42 ± 0.06 ^a^	0.47 ± 0.01 ^cd^	1.22 ± 0.01 ^b^	0.60 ± 0.08 ^cd^	−0.46 ± 0.01 ^g^
LTQ	−11.19 ± 0.29 ^i^	3.02 ± 0.29 ^b^	−1.42 ± 0.27 ^d^	−0.04 ± 0.08 ^bcd^	1.03 ± 0.05 ^a^	1.20 ± 0.35 ^b^	1.25 ± 0.24 ^ab^	−0.46 ± 0.15 ^g^
LYT	−13.37 ± 0.28 ^j^	3.63 ± 0.28 ^a^	−2.48 ± 0.33 ^e^	−0.03 ± 0.17 ^bcd^	0.93 ± 0.07 ^ab^	1.37 ± 0.38 ^ab^	1.36 ± 0.01 ^ab^	−0.52 ± 0.18 ^g^
PZX	−4.46 ± 0.19 ^h^	2.33 ± 0.19 ^c^	−0.32 ± 0.15 ^c^	0.24 ± 0.24 ^bc^	0.41 ± 0.08 ^cd^	−0.21 ± 0.13 ^e^	0.33 ± 0.06 ^de^	−1.94 ± 0.07 ^i^
PLE	−1.1 ± 0.09 ^c^	−0.62 ± 0.09 ^i^	0.21 ± 0.11 ^bc^	−0.03 ± 0.3 ^bcd^	0.33 ± 0.15 ^d^	0.63 ± 0.01 ^c^	−0.02 ± 0.05 ^f^	1.34 ± 0.07 ^ef^
PBC	−2.79 ± 0.21 ^d^	0.08 ± 0.21 ^gh^	0.37 ± 0.22 ^ab^	0.38 ± 0.52 ^b^	0.49 ± 0.31 ^cd^	0.75 ± 0.06 ^c^	0.32 ± 0.29 ^de^	1.73 ± 0.14 ^d^
PLW	−1.14 ± 0.04 ^c^	−0.96 ± 0.04 ^i^	0.16 ± 0.02 ^bc^	0.078 ± 0.04 ^bc^	0.47 ± 0.04 ^cd^	0.79 ± 0.05 ^c^	0.22 ± 0.28 ^ef^	1.44 ± 0.14 ^e^
PBC	−0.28 ± 0.3 ^b^	−2.35 ± 0.31 ^k^	−5.36 ± 0.59 ^f^	−0.68 ± 0.0 6^e^	−0.22 ± 0.08 ^e^	1.64 ± 0.45 ^a^	−0.69 ± 0.19 ^g^	6.15 ± 0.2 ^a^
HHB	−2.97 ± 0.02 ^de^	−0.78 ± 0.02 ^i^	−0.92 ± 0.50 ^d^	−0.22 ± 0.04 ^cde^	−0.17 ± 0.05 ^e^	0.71 ± 0.12 ^c^	0.15 ± 0.09 ^ef^	3.15 ± 0.04 ^c^
HXM1	0.53 ± 0.15 ^a^	0.20 ± 0.15 ^fgh^	0.1 ± 0.08 ^bc^	0.10 ± 0.06 ^bce^	0.39 ± 0.15 ^d^	−0.23 ± 0.04 ^e^	0.18 ± 0.21 ^ef^	−0.52 ± 0.13 ^g^
HXM2	−0.58 ± 0.13 ^b^	−1.44 ± 0.13 ^j^	−1.14 ± 0.28 ^d^	−0.44 ± 0.47 ^de^	0.56 ± 0.53 ^bcd^	0.88 ± 0.27 ^c^	0.31 ± 0.04 ^de^	3.23 ± 0.25 ^c^
HCY	−3.96 ± 0.04 ^g^	0.85 ± 0.04 ^de^	0.83 ± 0.48 ^a^	−0.45 ± 0.49 ^de^	0.58 ± 0.61 ^bcd^	0.28 ± 0.3 ^d^	0.71 ± 0.3 ^c^	−1.5 ± 0.12 ^h^
HDL	−3.21 ± 0.11 ^ef^	0.03 ± 0.12 ^h^	−2.44 ± 0.58 ^e^	−0.73 ± 0.08 ^e^	0.51 ± 0.12 ^bcd^	1.35 ± 0.1 ^b^	1.11 ± 0.02 ^b^	3.84 ± 0.08 ^b^

Different letters in Tables a~k indicate significant differences within the same column (*p* < 0.05).

## Data Availability

The original contributions presented in the study are included in the article, further inquiries can be directed to the corresponding author.

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
