# Peer review of "Effects of Different Production Methods on the Quality and Microbial Diversity of Sauerkraut in Northeast China"

_foods, 2024, doi:10.3390/foods13233947_

Round 1
Reviewer 1 Report
Comments and Suggestions for Authors
Manuscript with title "Effects of different production methods on the quality and microbial diversity of Sauerkraut in Northeast China" answer the question of sauerkraut production with three different methods very clearly. Comparison between technology approaches is valuable and original in complex of used methods and their combination. Work combine novel methods with microbial overview and analytical data. Authors comment data and compare different technologies. New information are clearly presented. Only small details can be cleared, as I mentioned, Nitrite content is not mentioned in methodology, but in discussion is relevant. In comparison with other published materials text work with real samples from the market, describe them in detail and compare. Only one possibility is to improve, but speculative. Is possible use this method for authentication of samples? Method against falsification of food is necessary also for fermented vegetable. Conclusions are consistent and references appropriate. Authors used for comparison information about sauerkraut adequately, other fermented foods are too different, cabbage is specific material. Text is informative and complex.
Author Response
Comments 1: Only small details can be cleared, as I mentioned, Nitrite content is not mentioned in methodology, but in discussion is relevant.
Response: Thank you for pointing this out. We agree with this comment. We have added the method to Section 2.2.
After the sample was incubated in a water bath for 15 min, it was mixed with saturated borax, and 106 g/L potassium ferrocyanide (Aladdin, USA) and 220 g/L zinc acetate (Aladdin, USA) were added to the solution and mixed evenly. After standing, the supernatant was mixed with 2 mL of 4 g/L p-aminobenzenesulfonic acid (Aladdin, USA) solution and 1 mL of 2 g/L N-(1-naphthalene) ethylenediamine hydrochloric acid (Aladdin, USA) for 20 min, and the absorbance was measured at a wavelength of 538 nm. The nitrite content was calculated on the basis of a standard curve. The nitrite content was expressed in mg/kg. (in line 109-116)
Comments 2: In comparison with other published materials text work with real samples from the market, describe them in detail and compare. Only one possibility is to improve, but speculative. Is possible use this method for authentication of samples? Method against falsification of food is necessary also for fermented vegetable.
Response: Thank you for pointing this out. We believe that it is feasible to identify sauerkraut from the Northeast (because literature shows that the bacterial profiles of sauerkraut from other regions may be quite different from that of the Northeast, which will lead to unstable results). Our experimental results show that there are differences in flavor and microbial diversity between different fermentation methods. Therefore, the production method of sauerkraut can be preliminarily judged based on the flavor substance composition and microbial composition of sauerkraut.
Reviewer 2 Report
Comments and Suggestions for Authors
The reviewed work that refers to the comparison between three types of sauerkraut of different origin is interesting. It is a complete work that provides valuable information about this food. I attach some observations to the work.
Lines 80-83. Indicate what the abbreviations mentioned mean.
Line 167. Why is the emphasis on nitrite content?
Lines 255-265. The reason for the presence of pathogens in sauerkraut is not fully justified. It is assumed that the addition of salt during the manufacture of the food favors the growth of halophilic microorganisms. Is this a manufacturing error?
Lines 236-250. What differences exist, specifically, in the manufacture of the food that favors or disfavors the presence of microorganisms?
Author Response
Comments1: Lines 80-83. Indicate what the abbreviations mentioned mean.
Response: Thank you for pointing this out. As described in Section 2.1, a total of 15 brands of sauerkraut were used in this study. The first five brands (LYY, LZY, LHX, LTQ, LYT) have the initials L, which represent fermentation by lactic acid bacteria, and the other letters are the abbreviations of the Chinese names of different brands (e.g., YY in LYY is the abbreviation of yuyuan). The first letters P of the middle five brands (PZX, PLE, PBCC, PLW, PBC) represent the addition of additives during the fermentation process, and the other letters are the abbreviations of the Chinese names of different brands. The first letters H of the last five brands (HHB, HXM1, 82 HXM2, HCY, HDL) represent homemade by farmers, and the other letters are the abbreviations of the Chinese names of different purchasing areas.
Comments 2: Line 167. Why is the emphasis on nitrite content?
Response: Thank you for pointing this out. I'm glad you read my manuscript so carefully. First, in the introduction (in line 51-55), it is described that the nitrite content of naturally fermented sauerkraut with additives will be reduced, and conversely, the nitrite content of naturally fermented sauerkraut without additives will be high. In addition, in other articles, the nitrite content of homemade sauerkraut is generally excessive, so the emphasis on nitrite content here is to highlight the particularity of the physical and chemical indicators of homemade sauerkraut in this article.
Comments 3: Lines 255-265. The reason for the presence of pathogens in sauerkraut is not fully justified. It is assumed that the addition of salt during the manufacture of the food favors the growth of halophilic microorganisms. Is this a manufacturing error?
Response: Thank you for pointing this out. We agree with this comment. In the introduction (inline 49-55), we mentioned some disadvantages of natural fermentation, which lead to different degrees of contamination in the production process of sauerkraut. Therefore, we think it is reasonable to find these pathogens in homemade sauerkraut. It is true that adding salt can lead to the growth of halophilic microorganisms, but in this study we focus on the effect of production methods on sauerkraut, and the effect of salt addition deserves to be studied separately.
Comments 4: Lines 236-250. What differences exist, specifically, in the manufacture of the food that favors or disfavors the presence of microorganisms?
Response: Thank you for pointing this out. We sincerely believe that your proposal is very valuable. We have added the issues you mentioned and marked them in red in the article.
" Lactobacillus are the main bacteria in the fermentation process of sauerkraut, which are related to the production of organic acids and the formation of flavor substances in sauerkraut. Its abundance has a significant impact on the quality of sauerkraut [28]. Lactic acid bacteria are directly added to sauerkraut as a fermentation agent, which directly increases the abundance of Lactobacillus, while the addition of additives may indirectly increase the abundance of Lactobacillus by reducing the abundance of harmful bacteria." (in line 272-278)
Reviewer 3 Report
Comments and Suggestions for Authors
GENERAL COMMENTS
The manuscript describes the effects of different production methods on the quality and microbial diversity of Sauerkraut in Northeast China.
From abstract
In general, it is an interesting description of the work, as a recommendation is necessary to describe in resume methodology before present results, and as the conclusion provides theoretical support for the production of high-quality Northeast sauerkraut based on which attributes.
The manuscript shows an adequate methodology, but it is missing the methodology for sample preparation in GC‒MS analysis, which only provided the operation condition of the equipment.
Line 109 “Tridecanoic acid methyl ester, at a concentration of 10 μg/mL, was utilized as an external standard for quantitative analysis”. This compound is an internal or external standard for quantification; this doubt is because sample preparation is missing,
For results
Line 175 does not present quantification or the analysis of volatile components and is based only on the presence or absence of relatives or their abundance.
In conclusion, a rewrite is proposed to clarify and be concise. There are three principal ideas: the grouping for different regions, the antimicrobial effect, and the development of compounds that improve flavor and odor and have been positively correlated with bacteria
Author Response
Comments 1: The manuscript shows an adequate methodology, but it is missing the methodology for sample preparation in GC‒MS analysis, which only provided the operation condition of the equipment.
Response: Thank you for pointing this out. We agree with this comment. We have added the issues you mentioned and marked them in red in the article.
Take 5 g of homogenized sauerkraut sample, place it in a headspace bottle, and place it in a water bath at 55°C for 30 minutes. After preheating, insert the SPME (solid-phase microextraction) fiber head and completely absorb it for 30 minutes, and then insert it into the GC injector at 250°C for 5 minutes. (in line 120-123)
Comments 2: Line 109 “Tridecanoic acid methyl ester, at a concentration of 10 μg/mL, was utilized as an external standard for quantitative analysis”. This compound is an internal or external standard for quantification; this doubt is because sample preparation is missing,
Response: Thank you for pointing this out. We agree with this comment. We have added the issues you mentioned and marked them in red in the article.(in line 120-123)
Comments 3: Line 175 does not present quantification or the analysis of volatile components and is based only on the presence or absence of relatives or their abundance.
Response: Thank you for pointing this out. We agree with this comment. After careful consideration, we believe that this sentence lacks evidence and is out of date, so we have deleted it.
Comments 4: In conclusion, a rewrite is proposed to clarify and be concise. There are three principal ideas: the grouping for different regions, the antimicrobial effect, and the development of compounds that improve flavor and odor and have been positively correlated with bacteria
Response: Thank you for pointing this out. We agree with this comment.We have rewritten or slightly modified several of the sections you mentioned
A total of 15 different brands of sauerkraut were purchased. According to their production methods, they were divided into lactic acid bacteria fermentation group Group-L (LYY, LZY, LHX, LTQ, LYT), additive group Group-P (PZX, PLE, PBCC, PLW, PBC), and farmer-made group Group-H (HHB, HXM1, HXM2, HCY, HDL). An appropriate amount of sauerkraut from each group was homogenized and stored at -20°C for subsequent determination of microbial diversity. The remaining samples were stored at 4°C until use. (in line 94-100)
This may be due to the unique metabolic processes of these bacteria, which produce substances that play a positive role in the flavor of sauerkraut.(in line 331-333)
Dehydroacetic acid was the most abundant acid in Group P. It is a substance with broad-spectrum antimicrobial effects. Its large presence may lead to a decrease in the diversity of Group-P microorganisms, which in turn affects the flavor of Group-P fermented sauerkraut, resulting in its poor flavor. (in line 341-345)
Reviewer 4 Report
Comments and Suggestions for Authors
The manuscript presents results very interesting, was good written and the approach used was sufficient for obtaining important insights concerning the sauerkraut production. I highlight the quality of the data analysis and its interpretation as a positive point of the paper. After correction, I found that only a few minor corrections are needed, as presented below:
· Figure 4 – Improve the quality, is difficult to observe the text in the axis of charts, especially in Figure 4A.
· Figure 6 – Improve the quality, is difficult to observe the text in the axis of charts.
· In References section, check the spelling of some names of microorganisms and correct them when necessary.
Author Response
Comments 1: Figure 4–Improve the quality, is difficult to observe the text in the axis of charts, especially in Figure 4A.
Response: Thank you for pointing this out. We agree with this comment. We have improved the quality and clarity of the images and enlarged the text to make them easier to read.
Comments 2: Figure 6–Improve the quality, is difficult to observe the text in the axis of charts.
Response: Thank you for pointing this out. We agree with this comment. We have improved the quality and clarity of the images and enlarged the text to make them easier to read.
Comments 3: In References section, check the spelling of some names of microorganisms and correct them when necessary.
Response: Thank you for pointing this out. We agree with this comment. We have corrected the spelling and format of some microbial names. Some of the same microbial names have different spellings (such as Lactobacillus Plantarum and Lactiplantibacillus Plantarum), so the spelling in the original title is still retained. (in line 413, 421, 460-461, 505, 518, 522)